# Universal health coverage—Exploring the what, how, and why using realist review

Aklilu Endalamaw[1,2*], Tesfaye Setegn Mengistu[1,2], Resham B. Khatri[1,3], Eskinder Wolka[4], Daniel Erku[1,5], Anteneh Zewdie[4], Yibeltal Assefa[1]

1 School of Public Health, The University of Queensland, Brisbane, Queensland, Australia, 2 College of Medicine and Health Sciences, Bahir Dar University, Bahir Dar, Ethiopia, 3 Health Social Science and Development Research Institute, Kathmandu, Nepal, 4 International Institute for Primary Health Care-Ethiopia, Addis Ababa, Ethiopia, 5 Centre for Applied Health Economics, School of Medicine, Griffith University, Brisbane, Queensland, Australia

* yaklilu12@gmail.com

## Abstract

Universal health coverage (UHC) is a critical target in many health system strategies to achieve 'good health and wellbeing'. Evidence on the meaning and scope of UHC and the strategies required to achieve it are needed, as variations in its understanding and implementation have risen. This realist review was conducted to fill the knowledge gap by synthesising evidence on the meaning, components, significance, and strategies of UHC. A review of evidence was conducted based on realist synthesis. We searched PubMed, EMBASE, Scopus, and Web of Science for published materials and websites for grey literature. We have followed some steps: define the scope of the review and develop initial programme theory, search for evidence, data extraction, and synthesise evidence. This review revealed that universal coverage, universal health, universal healthcare, universal access, and insurance coverage are used interchangeably with UHC. It is a legal notion that embodies a human rights-based and collaborative approach to ensure fair and comprehensive health care services. Universal health coverage is relevant for three macro reasons: first, it prevents and reduces the impact of diseases; second, it addresses inequality and promotes equity; and third, it is key for global health security. Various mechanisms are involved to implement UHC, such as health insurance, social health protection, digital financing systems, value-based care, private sectors, civil societies, partnerships, primary health care, and reciprocal health care systems. In conclusion, universal health coverage is a multifaceted concept that various terms can express in different contexts. Universal health coverage is a political and ethical imperative that aims to promote health equity and protect human dignity across different levels of society. It is essential in preventing diseases and crucial to global health security. Practically, UHC is not truly universal, as it does not include all services under its scheme and varies across countries. This requires consistent advocacy, strategic and operational research, and political will to ensure UHC.

**Data availability statement:** All available data have been included in the paper in the form of a summary table and supplementary files.

**Funding:** The authors received no specific funding for this work.

**Competing interests:** The authors have declared that no competing interests exist.

## Introduction

Universal health coverage (UHC) relies on key terminologies. 'Universal' represents 'covering all without limit or exception' [1]; 'health' as 'a state of complete physical, mental and social wellbeing and not merely the absence of disease or infirmity' [2]; and 'coverage' denotes 'something that covers' (health services and financial support) [3]. This concept has gradually become a prominent topic of discussion on the global stage [4]. Yet, the combined meaning of these three words seems to be aimed at achieving the 1978 primary health care (PHC) 'health for all' declaration; both concepts strive for a healthier world [5].

Following the Millennium Development Goals [6], the UHC agenda endeavours to increase access to essential health services and reduce financial catastrophe due to health services costs. There have been improvements towards UHC globally; however, half of the world's population still lacks access to essential health services [7]. In addition, approximately 2 billion people are currently facing financial hardships, with 1 billion experiencing catastrophic out-of-pocket health spending, and 344 million individuals being pushed further into poverty [8]. Less than 30% of countries (42 of 138 countries) were able to improve service coverage and reduce catastrophic out-of-pocket (OOP) health spending since 2015. This implies that the global progress towards UHC by 2030 is off-track [9]. Most countries (108 out of 138 countries) experienced worsening or no significant change in service coverage since 2015 [9].

Some of the factors behind this inadequate progress towards UHC include lack of political commitment, fragmentation between public and private sectors, and lack of comprehensive health policies, and weak health systems in low- and middle-income countries [10–12].

On the other hand, there is a lack of clarity about the meaning and scope of UHC. Universal health coverage has been misunderstood or misinterpreted in different countries [13,14]. For example, in Kenya, there was a lack of agreement on stakeholders' interpretation of what is to be understood by UHC [15]. This misunderstanding of UHC may create challenges and confusion in delivering services, monitoring progress, celebrating successes, and sustaining efforts towards UHC [16].

Knowledge and evidence on 'what works' to achieve UHC have been identified as critical facilitators [17]. Horton et al argued that 'the great gap that now exists for countries trying to deliver UHC is access to a library of knowledge to assist their decision-making' [18]. To address this gap, there has been an increasing number of research publications. These include exploration of mapping the available literature on UHC [12], identification of best practices towards realizing UHC [19], measurement and attainment of UHC for older people [20] and people with stroke [21], analysis of UHC's influence on health policy [4], and health financing of UHC [22]. Other publications include evolution and future challenges of UHC [23], and the processes and evidence of UHC [24]. These articles do not replace the current realist review, which synthesises all available evidence on UHC and addresses various perspectives from the literature, studies based on stakeholders' interview, editorials, and commentators. While some identified challenges and facilitators of UHC, they did not address the broader mechanisms or the necessity for launching UHC. There were variations in its understanding and implementation. Hence, it is timely to conduct this a realist review, which helps to inform UHC policy and implementation, highlighting its significance and mechanisms.

This article aims to synthesis the available evidence on the meaning of UHC, its components, its significance, and implementation strategies to achieve it.

## Methods

We conducted a realist synthesis of literature rooted in the UHC's definition, principles, and implementation based on Pawson and colleagues' realist review approach [25]. This realist

review approach follows four steps: define the scope of the review and develop initial programme theory, search for evidence, data extraction and appraisal, and synthesis of evidence.

## Clarify the scope of the review and existing definition

The realist review seeks to unpack the mechanisms by which strategies work and disclose for what reason or purpose it is required [26]. Implementing realist synthesis will allow a deep understanding of UHC and link its strategies with the target population in the community by realizing linkage as a mechanism [27]. We examined how, why, for whom and under what circumstances UHC works. As UHC strives to provide comprehensive services in health system, it is possible to introduce mechanisms that can support it [28]. The structured definition of UHC, as laid out by WHO in 2010, has three dimensions: population coverage, service coverage, and financial coverage [29]. Understanding the comprehensive pathway from the definition to the outcomes of UHC may require critical analysis and an understanding of the principles to answer what UHC means. Additionally, the review addresses how UHC operates and which mechanisms or enabling conditions. Mechanisms are processes for enabling conditions, which operates in specific contexts to achieve the outcomes (e.g., service coverage or financial protection).

## Search for evidence

The search strategy is considered to identify all relevant evidence on UHC. We conducted a broad search to identify articles on UHC. We conducted meetings and discussions on the detailed elaboration of the conceptual frameworks that underpin the key principles of UHC to guide the search processes in electronic databases, grey literature search engines, and institutional websites (WHO and the World Bank/WB/). The search strategy and keywords were based on a broad search to locate various concepts, definitions, comments, perspectives, and empirical results, which can provide an overview of the UHC. The broader search was supported by a focused and systematic search strategy and keywords for "Universal Health Coverage", "Universal Coverage", and "Universal Health Care". The search strategy was adapted to each electronic database. Boolean operator ('OR') was applied when necessary, depending on the database. S1 File provides a detail search strategy for databases.

We searched PubMed, SCOPUS, EMBASE, and Web of Science. Additionally, we accessed Google Scholar, Google, UNs, UNICEF, WHO and WB website to retrieve unpublished work, grey literature, and guidelines or appropriate reports. We also checked the reference lists of eligible articles for further inclusion. We included materials from the inception of each database up to the end of November 2023, time, settings or country, and article types. Articles contribute to one of the main addressed issues, which are what? how? and why? were considered, including editorials, opinion/position pieces, commentaries, realist evaluations, perspectives, qualitative and quantitative studies, programme manuals, and systematic and scoping reviews. The articles eligible for inclusion in the study were published in English.

We exported all retrieved articles to EndNote 20 and removed all the duplicate. We then screened articles using title and abstract screening, and full text using eligibility assessment criteria. The search results from Google Scholar, Google, UNs, UNICEF, WHO and WB websites, and searches from reference lists of those articles that passed full-text screening were again checked for duplication and final eligibility. After collecting the articles, we followed the PRISMA flow chart to select eligible articles and excluded those that were not relevant to UHC. The search strategy, the screening process, and overall activities were evaluated during weekly meetings of the authors of this review.

### Appraise and extract evidence

Article appraisal in realist review does not follow a similar approach to traditional systematic review, which categorizes articles in a hierarchy. Realist review examines complex areas of reality by considering highly diverse sources of evidence. It relies on specific elements of reports from the included article to support context, mechanism, and outcome. Hence, in a realist review, appraisal of evidence provides attention to ensure whether an article addresses the issues under examination or the worth of an article in evidence synthesis. Data extraction is followed after deciding to include relevant articles in the evidence synthesis [25].

Data extraction in this realist synthesis was conducted by taking notes, highlighting, and labelling the relevant sentences that fit what UHC mean and how does it work (action, mechanism, and outcome) Or why UHC is important? This is based on the realist review data extraction approach by which 'realist review assimilates information more by note-taking and annotation than by extracting data as such'[25]. In this review, data extraction was conducted first and then data relevant to how UHC operates, which consists of action, mechanisms, and outcome, was checked. Finally, relevant information on the importance of UHC was highlighted and labelled. One article was evaluated for these three areas before being passed to the next article. However, it was not necessary for an article to present the three main areas; it was enough if it included only one of the three questions. Overall, an iterative data extraction process was conducted.

### Evidence synthesis

Synthesis of the extracted data was conducted according to the realist review approach [25]. In this phase, the extracted data, highlighted during the data extraction phase, was collated into a similar or unique theme. Themes were developed under the what, how, and why of UHC (according to the current review questions) based on the findings from the included evidence. The synthesis was mainly dependent on reasoning using a realist logic of analysis. We also presented Australia's experience as a case study primarily to show the three dimensions of UHC and its supportive strategy. The data for this case study was based on the date from the Australian Government website as of 25 December 2023 [30].

## Results

### Search results

A total of 25,853 documents were found in electronic databases and websites. A total of 12,148 were found in Scopus, followed by PubMed (8,426), Web of Science (2,658), EMBASE (2,542), and Google Scholar (71). Additionally, we have found eight documents from websites (UN, WHO, WB, and UNICEF) (Fig 1). A total of 124 articles were included in this review. Characteristics of included documents, such as author/year, article category, study design, country, and inclusion purpose, are provided in S2 File.

### Main findings

The findings of the review are presented below according to the objectives of the study: (1) what is universal health coverage? (2) why is universal health coverage relevant? and (3) how is universal health coverage implemented?

### What is Universal Health Coverage?

Our synthesis shows that UHC is expressed interchangeably with universal coverage, universal health, universal healthcare, universal access, and insurance coverage [31–38]. Likewise, operational managers in South Africa defined UHC as 'health for all' [39].

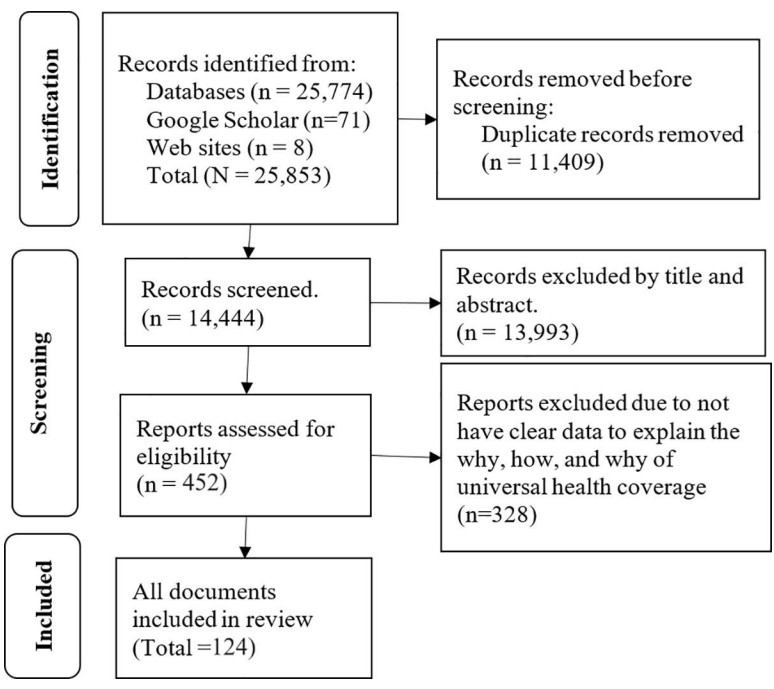

**Fig 1. Article selection process.**

In some context (e.g., Asia and the pacific region), UHC is also explained as a process of delivering quality, equitable, and comprehensive care and services, including promotive, preventive, curative, rehabilitative, and palliative care with affordable cost or without financial barriers [40,41]. Here, it is important to note that the term 'affordability' of care has various understandings, including free of costs (e.g., avoiding direct payments), prepaid through government-funded health service scheme, having large risk pools to ensure sustainable cross-subsidy between the healthy and wealthy and the sick and the poor, and covering the informal sector and households living on or below the poverty line financially [40–42]. Under UHC, people can access services regardless of their ability to pay, and no citizen is denied service due to inability to pay [10].

Universal health coverage is considered as legal concept and humanitarian concept, social (equitable entitlement and enrolment to the social security system), a health economic (financial protection), and a public health concept (comprehensive services versus essential packages) [43]. In these contexts, our synthesis shows that UHC has critical values, such as the right to health, social justice or equity in access, solidarity, quality, service coverage, population coverage (inclusiveness), and financial protection (affordability) [9,19,32,44–47]. Policy actors also identify values, such as quality of care, equity, financial risk protection, and a comprehensive set of services for every one with any health conditions as features of UHC [10].

Universal health coverage is defined as a target or goal: access and financial protection [9]. In this circumstance, UHC should be evaluated and measured for 'what services are covered?', 'how much most people pay to receive services?', and 'who is covered?' Globally, UHC is monitored using effective service coverage and financial protection indicators [44].

Universal health coverage regarding service coverage is called effective service coverage in the monitoring metrics. Effective coverage is a measure that unites intervention need, use, and

quality [48]. However, it is important to note that UHC is beyond the UHC effective service coverage index, providing a comprehensive set of services [39,49,50]. The UHC effective service coverage encompasses essential services or health benefit packages (HBPs) prioritized for evaluation and inclusion within the UHC scheme [51], which have been developed based on certain criteria [52–55].

Globally, UHC effective service coverage index has been estimated using tracer indicators, derived from NCDs; infectious diseases; Reproductive, Maternal, Newborn, and Child Health; and service capacity & access [9,56–61]. Across different countries researchers have developed UHC monitoring indicators for specific country contexts. For example, governance, stewardship and financing, and palliative care are added indicators besides the fundamental tracers indicators in India [61]). Similarly, the sustainability of healthcare, need cost, health outcomes, enabling the environment for benefitting from healthcare, quality of healthcare, and human resources for health are served as UHC monitoring indicators from supply-and demand-sides of health services [62].

Other fundamental issues for the question 'Which services are covered?' and 'Which services are not covered? can be viewed from the services covered by health insurance. For example, in developed countries, eye services have been delivered as a component of UHC [63], while oral health care is not publicly covered [64,65].

Financial protection matters about the amount that people pay to receive services. Financial protection mainly stands for insurance coverage or subsidized care. In this dimension, the UHC-related questions are: 'who is covered (eligible population for and percentage of population covered by financial support)?', 'what is covered (service package covered and drugs covered by financial support)?', and 'how much is covered (per capita premium and government subsidy per capita)?'[66]. Financial coverage covers livelihood expenses, economic assistance, and reducing medical costs [50,67]. Financial protection and/or affordability are measured or monitored as part of UHC [59,60]. Globally utilized financial protection monitoring indicators, as part of UHC, include out-of-pocket (OOP) health spending, catastrophic health expenditure, impoverishment, and general financial hardship [9,67]. To illustrate, the percentage of households experienced catastrophic health expenditures at the threshold of 10% or 25% of total expenditures or 40% of non-food expenditures, and impoverishment, including the percentage of households pushed into poverty by OOP [60,68–72]. It is also measured by the poverty gap due to OOP payments, as described by the average amount of expenditures falling below the poverty line for those impoverished [60,71].

Population coverage, as parts of UHC, refers to who is covered for services and financial supports. Universal health coverage is designed to provide need-based quality and affordable health services across the life course for all groups. Sometimes, population coverage reflects the coverage of services for specific age categories and vulnerable groups [50]. For instance, the WHO and WB monitoring indicators show different indicators for groups of population and healthcare service types or diseases conditions. To illustrate, women of reproductive age (15–49 years) for family planning and antenatal care, infants for diphtheria-tetanus-pertussis containing vaccine, children under five years of age for care of suspected pneumonia, people living with HIV for antiretroviral therapy, population in malaria-endemic areas for insecticide-treated net access, households using at least basic sanitation facilities (all people live in the house), adults aged 30–79 years for prevalence of hypertension, and adults aged over 15 years for tobacco use [9]. Similarly, GBD-19 estimate classified population coverage into six groups: women 15 to 49 years, newborns, children younger than 5 years, children and adolescents aged 5 to 19 years, adults aged 20 to 64, and older adults aged 65 years or older [57].

## How does universal health coverage work?

The typical implementation mechanisms that support UHC include health insurance, a social health justice policy, digital financing systems, and partnerships (Table 1).

Healthcare insurance: Providing financial support is a strategy to cover all people in need towards universal coverage and providing quality- and low-cost care to prevent catastrophic expenditure, impoverishment, and poverty [73,74]. Healthcare insurance has several forms, such as Medicaid and Medicare [75], social health insurance [75,76], and private health

**Table 1. The contexts, strategies, mechanisms, and outcomes in implementing UHC.**

| Contexts and strategies | | Mechanisms | Outcomes |
|---|---|---|---|
| Health system | Primary health care | • Effectiveness, comprehensiveness of care and interpersonal quality of care<br>• Positive discrimination against the underserved population | UHC effective service coverage |
| | Value-Based Health Care | • Skill mix | UHC access and health outcome |
| | Digital system | • Make the system more responsive.<br>• Enabling programs to implement changes based on client-proposed features.<br>• Improve access to care<br>• Improve diagnostics and resource allocation | UHC: access and quality of care |
| | Health promotion | • Reduce hospital admission.<br>• Reduce demand for expensive health service. | Effective use of health resources |
| Health insurance | National health insurance | • Automatic coverage of all citizens via tax funds<br>• Use of tax revenues to subsidise target populations, steps towards broader risk pools, and emphasis on purchasing services through demand-side financing mechanisms | UHC: access & Financial protection |
| | Private health insurance | • Provide subsidized care for privately registered clients | Prevent CHE |
| | Social health insurance | • Funded via income-based contributions that require tax-based revenue. | Financial protection |
| Social health justice policy | Social protection | • Employment to unemployed<br>• Support the living cost.<br>• Prevent poverty<br>• Support people under the poverty line | UHC: access & prevent CHE |
| Partnership | Reciprocal health care agreement | • Visitors access care without paying from their pocket | UHC: access & prevent CHE |
| | (Inter)national partnership | • Expanding access to essential medicines and health products<br>• Technology transfer<br>• Develop an adequate health workforce | UHC: access |
| Voluntary organizations | Civil society | • Connect with difficult-to-reach populations.<br>• Achieve wider coverage and contribute to minimizing costs.<br>• Supplementing government emergency aid response<br>• Delivering necessary resources and supporting highly vulnerable populations | Access and financial protection |
| Private sector | Private sector | • Innovations for health care delivery<br>• Networks for extending service coverage and access to medicines and other medical commodities and financial protection.<br>• Transferring know-how for the maintenance and use of the equipment<br>• Pool different types of funding, | UHC and health security |
| Political interest | Consider as global public health and political agenda | • Declare UHC day<br>• Use as the election campaign<br>• Consider legislation and constitution documents | UHC |
| Addressing social determinants of health | Governance index, stability index and socio-demographic index | • Increased coverage of health service indicators | UHC |

insurance [77]. To illustrate one by one, single-payer or national health insurance (e.g., Medicare) is a form of free government-owned healthcare provided to everyone. At the same time, citizens should pay for income tax revenue [75,78–84]. There are also mixed health plans, such as applying Medicare as a public health insurance and purchasing private hospital coverage or paying a tax surcharge as subsidized private care [85,86]. Social health insurance is funded via income-based contributions that require tax-based revenue, though its application may vary depending on countries [75,76,87]. In developing countries, people voluntarily pay for health insurance as a prepayment so that they are exempted from paying medical costs, like initiatives of community-based health insurance scheme in Ethiopia [88] and Senegal [89]. In some instance, insurance schemes have been implemented for specific groups of the population, such as National Disability Insurance Scheme in Australia [90].

Social protection: It expands service coverage and prevents financial protection [91,92]. For example, social health protection initiatives (Sehat Sahulat Program in India) enrol the poorest population and covers all secondary and limited tertiary services with a maximum Indian Rupee 540,000 expenditure limit per family per year (the government pays a premium of Indian Rupee 1549 per year per household to 3rd party), which ensures services through a mix of public-private providers [93]. 'Seguro popular' is another example of social protection scheme that closes the gap in financial coverage and enhances comprehensive health service accessibility in Mexico [94].

Value-Based Health Care: It is a development of a health workforce with a mix of skills, and alternative funding models other than fee-for-service that initiated in Victoria, Australia aimed at reforming oral health care services [95,96].

Digital system (e.g., telemedicine and artificial intelligence): It contributed to UHC by making the system more responsive, enabling programs to implement changes based on client-proposed features, and improving healthcare access, diagnostics, resource allocation, and quality of services [75,97–100].

Strengthening PHC: PHC expenditure positively effects UHC with the involvement of the household and community, at which level PHC expands services and strategies [101–103]. PHC strengthens the healthcare system, empowers the community, and provides multisectoral action and policy [104].

Civil society: It helps the health system to connect with difficult-to-reach or vulnerable populations, supplementing government emergency aid response, and delivering necessary resources [105–107].

Private sector: Private sector and community engagement alongside the public health sector creates and fosters efforts and mobilizes resources towards UHC, health security, and sustainable development goals [108,109].

Social innovation: Social innovation facilitates achieving health through community co-learning, leadership, and accountability, providing the communities with respect and the opportunity to participate equally in creating and implementing programs by which communities were empowered, conceptualized, executed, monitored, and sustained the social innovation initiatives [110].

Reciprocal health care agreement: People, mainly visitors, from other countries, are eligible to receive subsidized medical costs [111].

Health promotion: Health promotion, by itself, is part of UHC. In another way, it has the power to reduce hospital admission and healthcare costs, resulting in the effective use of health care resources [112].

Partnership: This is about with-in-a-nation and cross-nations partnerships. Partnerships between government employers and consumers within a nation strengthen infrastructure capacity, raise funds for UHC, and provide better and comprehensive care towards UHC

[89,113,114]. As a cross nations partnership, the partnership project between Thailand and Japan [115] and China-Africa health development initiatives [116] towards UHC can be examples.

Political interest: Universal health coverage is a global public health and political agenda. Pursuing of UHC is inherently a political process [4,41,117,118]. United Nations member states have strongly recommitted to achieving UHC [119], and have declared December 12th as UHC Day. They urge all stakeholders, from its Member States and organizations to civil society and individuals, to celebrate it yearly and promote UHC's importance [120]. Universal health coverage has been utilized as a platform for a political election campaign. For instance, in India, the ruling centre-right government introduced the 'Pradhan Mantri Jan Arogya Yojana' reform shortly before the national election [121]. The political attention garnered by UHC has led to a constitutional amendment in Mexico [122]. The global movement towards UHC [123], the adoption of UHC as a policy reflecting legislative or executive commitment [124], and political motivation or commitment [109,125–127] are among the motives for implementing UHC.

Addressing social determinants of health: Structural determinants, such as the governance index, stability index and socio-demographic index [128], and growth domestic product and health expenditure [129] are positively associated with UHC. Mothers' higher education status is a factor for increased coverage of health service indicators. In contrast, the higher number of children and elderly population in the households were potential risk factors for an increased risk of catastrophic and impoverishing health payments in Iraq [130]. In general, the contexts, mechanisms, and outcomes underpinning the practice of UHC are presented in Table 1.

Presenting a country's experience with UHC is essential, particularly focusing on its implementation, the inclusivity of coverage, the extent of services, and the identification of those not covered. Additionally, understanding the supporting programs that enhance UHC in a specific country is vital. The readily available public information on UHC in Australia allows straightforward incorporation as a case study in this realist review. Australia's Medicare, a universal health insurance scheme, covers a wide range of services for citizens, permanent residents, and certain temporary residents. It includes hospital services, medical consultations, mental health services, some dental procedures, imaging, scans, pathology tests, eye tests, and medicines. However, it doesn't cover ambulance services, most dental services, glasses, contact lenses, hearing devices, elective and cosmetic surgery, and services not on the Medicare Benefits Scheme list. It is funded through income tax, including the Medicare levy and surcharge (S3 File) [30].

## Why Universal Health Coverage?

There are multiple compelling reasons to advocate for UHC. The primary objective of UHC is to foster a healthier population by preventing diseases and mitigating their consequences. It offers protection against epidemics, disasters, and other health crises. Additionally, it contributes to advancing gender equality and human rights.

## Mitigating the burden of diseases

Universal health coverage is instrumental in improving health outcomes, population health, and life-expectancy [131–134]. For example, in south east Asia and the Western Pacific region, countries with higher UHC scores experienced a significantly lower number of infected patients and deaths due to COVID-19 [135]. A single-payer UHC system is estimated to save numerous lives in the USA [136]. Additionally, a significant relationship was observed between

the UHC effective service coverage index and the burden of emergency diseases, substantially reducing disability-adjusted life years of emergency medical conditions [137]. Implementing UHC can also prevent antimicrobial resistance by reducing self-medication [138].

### Achieving equity

Universal health coverage focuses on equity, aiming to reach the left behind individuals [139,140]. The UHC approach expands essential healthcare services to the poorest [141,142], through which health equity can be achieved [143–146]. Universal health coverage is also linked with other sustainable development goals, exemplified by the call for UHC action to address gender inequities [147]. Social determinants of health and UHC have a bidirectional relationship.

### Health security

Universal health coverage contributes substantially to global health security (GHS). Global health security and UHC share common goals of preventing risk and fostering human rights, underscored by shared indicators such as health security, access to medicines, health work-force, and financial protection [148]. Global health security and UHC synergies support effective pandemic preparedness and response [149]. Both global health strategies have influenced each other. Universal health coverage indices, particularly maternal and child health and infectious diseases indicators, correlated with global health security [150]. Earthquakes adversely affected women's access to quality essential health services, reflecting the influence of GHS on UHC [151]. Furthermore, international health regulation core capacity scores were associated with the UHC service coverage index [152].

### Discussion

In this realist review, we explored and synthesised what, why, and how of UHC. Our analysis observed that the UHC has been utilized as a policy, process, and target to improve health. Universal health coverage comprises a collection of legal and social ideas whose foundation is laid by (inter)national organizations with the involvement of stakeholders. Likewise, the broad context of UHC addresses the needs of individuals based on national and global agreements. It is acceptable to stakeholders and beneficiaries due to its responsive values such as human rights, equity, justice, and quality. Hence, UHC fulfills health policy's features, like relevance, acceptability, effectiveness, efficiency, and equity.

In the context of the service provision process, UHC conceptualized as providing a range of health services to achieve a planned national and global target embedded with several non-mutually exclusive interdependent tasks. The United Nations member states aspire to UHC through the interaction between health, finance, economic, social, and political sectors. This implies a multisector interaction and agreement manifested by overt actions passing through a process. The actions in the processes are activities on equity in access, verifying human rights and justice, and ensuring financial protection, leading to maximizing UHC's effective service coverage and financial protection index. These indices are targets that are not overnight tasks; instead, they require the implementation of activities and strategies for more than 15 years as planned globally. As a target, the UHC effective service coverage and financial protection index have varied across countries because different countries may adopt various approaches depending on their context and need, indicating the contexts may be varied in implementation.

Health insurance is a prominent context-based strategy every country needs to achieve UHC. Its immediate and proximal benefits are to protect an individual from financial

catastrophe due to health care utilization costs and ultimately reduce CHE, poverty due to OOP expenditure, and impoverishment. The financial protection principle does not mean healthcare should be free of charge; instead, it addresses what charges patients should incur for using the services. Various types of health insurance across countries may differ based on the mechanisms of payment system for health care, the proportion of cost covered by direct pocket from patients and government, and the extent of coverage. Different health care models have been adopted based on the costs covered for health care services. For example, the Beveridge model provides health care for all citizens and is financed by the government through tax payments. This model is applied in Great Britain, Spain, and New Zealand [153]. Similarly, national public health insurance has been implemented in Australia. As of January 2024, all health care services except dental services for adults and theatre and hotel reservations from private health care services were covered in Australia [30].

In contrast, in a less developed country, such as Ethiopia, voluntary community-based health insurance (CBHI) (annual prepayment by people as a membership) and social health insurance (3% of the gross salary paid by volunteer employees and pensioners for themselves and their families) are the main options. In both insurance schemes, in Ethiopia, there were no clear directions about the services covered and not covered and the percentage of costs covered by patients and insurance schemes, except that it was described that essential health care services were covered based on the agreement and that the cost of health services for any beneficiary who used the health service without following the referral system would not be covered by the CBHI [154,155]. It is reported that appropriate government health expenditures have achieved a better reduction of infant mortality and an increase in the population's life-expectancy [156], and the implementation of public insurance coverage has led to an increase in inpatient and outpatient care utilization [157]. Among middle-income countries, Thailand has implemented a tax-financed scheme to provide free healthcare at the point of services, offering a comprehensive benefit package focusing on primary care [158]. Likewise, the 'Universal Access with Explicit Guarantees' initiative, which started in 2005, supports the achievement of UHC in Chile [159]. Health insurance-related moral behaviour should be carefully monitored because UHC in Indonesia has been associated with increased unnecessary admissions and readmissions [160].

Another macro context that affects UHC is partnership, which refers to the broader political, policy and cultural environment that influences how partnerships are formed, governed, and implemented. The partnership can create opportunities for UHC's progress. For example, China has made remarkable contributions to Africa's health development by building public health facilities and systems, enhancing basic medical service capacity, and improving human resources for medical services [161]. Reciprocal health care is a form of partnership between countries as an international agreement. For example, Australia has reciprocal health care agreements with 11 countries that enable many Australian travellers to access health care in other countries without paying out of pocket [162,163]. A public-private sector partnership is another way by which the UHC can be achieved, although it needs careful considerations that particularly involve the private-for-profit sector in the provision of health services due to their urban-biased distribution and the quality and pricing of their services [164].

On the other hand, the public sector sometimes neglects the availability and provision of some services that the private sector does, like basic obstetric and emergency care, which is neglected in the public sector [165]. Thus, public-private partnership creates synergies of power and complement each other. Moreover, a global partnership is not always practical if each member's interest deviates from the common agenda [166], ministries have weak ownership of policy dialogue, and stakeholders lack of confidence in their capacity for joint action [167]. Overall, a partnership might consider the national and local objectives, legal

and regulatory systems, financial and accountability arrangements, influence and interests of various stakeholders, and values and norms that influence collaborative work.

In general, several strategies have directly fostered UHC, and some have had a mutual interaction with UHC. For instance, value-based health care is among the strategies rarely presented in supporting UHC implementation. On the other hand, UHC can improve the implementation of value-based health care. It is understandable that the ultimate goal of value-based health care is to achieve health outcomes, improve client experiences of healthcare, and reduce costs to healthcare systems by reshuffling the existing service provision pathways [168]. The interdependent pillars of value-based health care include 'Organize into integrated practice units', 'measure outcomes and costs for every client', 'move to bundled payment for care cycles', 'integrate care delivery across separate facilities', 'expand excellent services across geography', and 'build an enabling information technology platform' [169]. Both intersect on health outcomes, healthcare costs, and expanding services. This indicates that activities to enhance these would serve twofold purposes: improving value-based healthcare and UHC.

The ultimate reason why UHC has gained a place on the global stage and in each country's national plan is to create a healthier society. Regarding the political economy of UHC, it is a political interest driven by various social forces to create public programmes or regulations that expand access to care, improve equity, and pool financial risks [41]. Unless properly managed, the political context, on the other hand, can hinder the financing of surgical care for children [118]. Moreover, social determinants of health and UHC have a bidirectional relationship. Addressing the challenges related to social classes can enhance progress towards UHC. Conversely, UHC is essential to addressing social determinants of health, improving health security, mitigating the consequences of health problems, and achieving health equity. The progress towards UHC can address the conditions in which people are born, grow, live, and conduct daily life, targeting social determinants of health [140].

Regarding implications, healthcare institutions and health education centers play a role in translating the concepts and purposes of UHC. Adopting effective mechanisms that foster UHC and enhancing local population engagement in planning and implementation is essential, ensuring that services are culturally acceptable and appropriate. It is also crucial to keep monitoring and evaluation activities on the implementation status of UHC initiatives, identify weaknesses and challenges, and align broader concepts, programs, and policies with social and economic strategies to tackle the social determinants of health, ensure health security, and achieve health equity.

### Limitation of this review

The review included only articles published in English.

### Conclusions

Universal health coverage is a multifaceted concept that can be expressed in various terms depending on the context and perspective. UHC is a political and ethical imperative that aims to promote health equity and protect human dignity across different sectors and levels of society. Practically, UHC is not truly universal, as it does not include all services under its scheme and varies across countries. Investing in and implementing the various mechanisms that can facilitate UHC for all is imperative. Due to political supremacy, preventing diseases and mitigating their effects, addressing socioeconomic determinants of health to attain equity, and serving a crucial role in global health security, universal health coverage is essential for all nations.

## Supporting information

**S1 File. Search strategies.**
(DOCX)

**S2 File. Characteristics of articles.**
(DOCX)

**S3 File. Australia's Public Health Insurance system as a case study for UHC.**
(DOCX)

**S1 Checklist. PRISMA checklist.**
(DOCX)

## Author contributions

**Conceptualization:** Aklilu Endalamaw, Yibeltal Assefa.

**Data curation:** Aklilu Endalamaw, Yibeltal Assefa.

**Formal analysis:** Aklilu Endalamaw.

**Investigation:** Aklilu Endalamaw.

**Methodology:** Aklilu Endalamaw.

**Supervision:** Aklilu Endalamaw, Yibeltal Assefa.

**Validation:** Aklilu Endalamaw, Tesfaye Setegn Mengistu, Resham B. Khatri, Eskinder Wolka, Daniel Erku, Anteneh Zewdie, Yibeltal Assefa.

**Visualization:** Aklilu Endalamaw, Tesfaye Setegn Mengistu, Resham B. Khatri, Eskinder Wolka, Daniel Erku, Anteneh Zewdie, Yibeltal Assefa.

**Writing – original draft:** Aklilu Endalamaw.

**Writing – review & editing:** Aklilu Endalamaw, Tesfaye Setegn Mengistu, Resham B. Khatri, Eskinder Wolka, Daniel Erku, Anteneh Zewdie, Yibeltal Assefa.

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
