## [Decision Letter · Decision Letter 0]

1 Oct 2024

PGPH-D-24-01157

Universal Health Coverage – Exploring the What, How, and Why Using Realistic Review

Dear Dr. Endalamaw,

Thank you for submitting your manuscript to PLOS Global Public Health. After careful consideration, we feel that it has merit but does not fully meet PLOS Global Public Health’s publication criteria as it currently stands. Therefore, we invite you to submit a revised version of the manuscript that addresses the points raised during the review process.

Your manuscript has been assessed by three reviewers and their comments are available below. They have requested clarity on the search strategy and improvements in the reporting of the results. Please review their comments and make the appropriate revisions. 

We look forward to receiving your revised manuscript.

Kind regards,

Emma Campbell, Ph.D

Staff Editor

Journal Requirements:

1. We would like to request for copy editing

Reviewers' comments:

Reviewer's Responses to Questions

**Comments to the Author**

1. Does this manuscript meet PLOS Global Public Health’s publication criteria ? Is the manuscript technically sound, and do the data support the conclusions? The manuscript must describe methodologically and ethically rigorous research with conclusions that are appropriately drawn based on the data presented.

Reviewer #1: Yes

Reviewer #2: Yes

Reviewer #3: Yes

2. Has the statistical analysis been performed appropriately and rigorously?

Reviewer #1: N/A

Reviewer #2: Yes

Reviewer #3: N/A

3. Have the authors made all data underlying the findings in their manuscript fully available (please refer to the Data Availability Statement at the start of the manuscript PDF file)?

Reviewer #1: Yes

Reviewer #2: Yes

Reviewer #3: Yes

4. Is the manuscript presented in an intelligible fashion and written in standard English?

Reviewer #1: Yes

Reviewer #2: Yes

Reviewer #3: Yes

5. Review Comments to the Author

Reviewer #1: The article is too descriptive, and contents in the main findings can be shorten, and write more concisely than the submitted manuscript. The manuscript looks like a chapter of Master degree or PhD thesis. It is unclear about the screening process to exclude a number of literature (n=13,993) using the title and abstract of the records. In addition, the discussion section should be written concisely and clearer than the draft manuscript.

Reviewer #2: Revise based on reviewer comments

Reviewer XXX

PLOS GLOBAL PUBLIC HEALTH

Universal Health Coverage – Exploring the What, How, and Why Using Realistic

Review

Good areas

The paper is interesting reading about Ethiopian community-based health insurance and the issues of reluctance to renew their membership to the community-based scheme. I have learnt and gain some useful insights on the study.

The discussion section is well done and its far better than the presentation of findings

Areas that need some clarity and revision

It is not clear if the two words or areas mean the same in the paper.

1. The main purpose and the problem of investigation should be stated clearly in the abstract and in the introduction.

2. In the title it is realistic review, in the keywords, it is realist review. The authors may clarify the two terms or words

3. I am not sure if the authors used systematic review or not, also if PRISMA was part of their search for secondary data. As stated on page 5: “We searched PubMed, SCOPUS, EMBASE, and Web of Science. Additionally, we accessed Google Scholar, Google, WHO website, UNs, UNIECF, and the WB website to retrieve unpublished work, grey literature, and guidelines or appropriate reports. We also checked the reference lists of eligible articles for further inclusion”.

4. These words should not be Bold from pages 12-13 example Value-Based Health Care…

5. The study findings are loosely presented, and I must say the findings are not well-done.

I suggest the authors present the findings along some major themes rather that the multiple sub-themes, which are too clumsy. The style of presentation of the findings should be revised and presented with some major themes.

Reviews XXX Recommendation

Accept submit to revision

September 2024

Reviewer #3: The paper does a good job of bringing together and summarizing the meaning and scope of Universal health coverage and the strategies required to achieve it. UHC is easy to say but hard to implement comprehensively. This paper reviews the meaning, components, significance, and strategies of UHC, which will help researchers and policymakers better understand it. Some revision should be taken in this paper.

Introduction

- Some sentences are repeated: (This will provide detailed understanding of UHC, the reasons and significance and relevance of UHC, as well as the mechanisms towards UHC.)/(This article aims to synthesis the evidence on the meaning of UHC, its components, its significance, and implementation strategies to achieve it..),

- There is incoherence between the claims made in the introductory of the paper and the main take that I derived. For example, in Paragraph 5 the authors claim that ‘We argue that it is timely to systematically and realistically review and synthesize the available evidence to guide policy and implementation of the UHC agenda.’ but the implementation strategies and action plans are not explicitly spelt out in the paper. This is very important, because health systems in different contexts are at different maturity levels pursuing different objectives related to their priorities.

-The Introduction section could be made stronger by acknowledging health financing. For example, in Paragraph 5, other than the mentioned research publications like “measurement and attainment of UHC for older people and people with stroke, there are a number of research publications on health financing of UHC.

Method:

- As there are few published examples of realist synthesis, please verify your reason for conducting a realist synthesis and discuss its focus.

-In “Search for evidence” section, the authors claim that they have included materials without restriction on language. But, in “Limitation of this review section”, they mention that only articles published in English have been included.

Discussion

It would be helpful to cast light on the UHC in middle- income countries. The countries selected to be discussed are either well- developed or less developed countries.

Minor comments

-Consider introducing the policy implications section in the discussion

- Please write the full word for these abbreviations: RMNCH and CHE. (Reproductive, Maternal, Newborn, and Child Health)/ (Catastrophic Health Expenditures).

6. PLOS authors have the option to publish the peer review history of their article (what does this mean? ). If published, this will include your full peer review and any attached files.

**Do you want your identity to be public for this peer review?** For information about this choice, including consent withdrawal, please see our Privacy Policy .

Reviewer #1: No

Reviewer #2: **Yes: ** Daniel Dramani Kipo-Sunyehzi

Reviewer #3: No

---

## [Decision Letter · Decision Letter 1]

19 Nov 2024

Universal Health Coverage – Exploring the What, How, and Why Using Realist Review

PGPH-D-24-01157R1

Dear Mr. Endalamaw,

We are pleased to inform you that your manuscript 'Universal Health Coverage – Exploring the What, How, and Why Using Realist Review' has been provisionally accepted for publication in PLOS Global Public Health.

Best regards,

Rajesh Sharma, Ph.D.

Academic Editor

Reviewer Comments (if any, and for reference):

Reviewer's Responses to Questions

**Comments to the Author**

1. If the authors have adequately addressed your comments raised in a previous round of review and you feel that this manuscript is now acceptable for publication, you may indicate that here to bypass the “Comments to the Author” section, enter your conflict of interest statement in the “Confidential to Editor” section, and submit your "Accept" recommendation.

Reviewer #1: All comments have been addressed

Reviewer #3: All comments have been addressed

2. Does this manuscript meet PLOS Global Public Health’s publication criteria ? Is the manuscript technically sound, and do the data support the conclusions? The manuscript must describe methodologically and ethically rigorous research with conclusions that are appropriately drawn based on the data presented.

Reviewer #1: Yes

Reviewer #3: Yes

3. Has the statistical analysis been performed appropriately and rigorously?

Reviewer #1: N/A

Reviewer #3: N/A

4. Have the authors made all data underlying the findings in their manuscript fully available (please refer to the Data Availability Statement at the start of the manuscript PDF file)?

Reviewer #1: Yes

Reviewer #3: Yes

5. Is the manuscript presented in an intelligible fashion and written in standard English?

Reviewer #1: Yes

Reviewer #3: Yes

6. Review Comments to the Author

Reviewer #1: The authors can address all comments from the three reviewers and also shorten the manuscript to be more concise than the previous version. The revised manuscript is well written and can explain clearly about the findings related to definition of UHC and key success of implementation strategies through the realist reviews. We do believe that readers will gain new knowledge on what, how and why countries in different regions and settings implement the policy on universal health coverage in different contexts and circumstance. In addition, the key success factors and lessons learned from different countries in different contexts would be more understood by the readers.

Reviewer #3: The authors adequately addressed the comments made by the reviewers and also my feedback from the first round of peer review. Therefore, I have no further comments

7. PLOS authors have the option to publish the peer review history of their article (what does this mean? ). If published, this will include your full peer review and any attached files.

**Do you want your identity to be public for this peer review?** For information about this choice, including consent withdrawal, please see our Privacy Policy .

Reviewer #1: **Yes: ** Dr Phusit Prakongsai

Reviewer #3: No
